# Risk Factors for Severe Postoperative Complications after Oncologic Right Colectomy: Unicenter Analysis

**DOI:** 10.3390/medicina58111598

**Published:** 2022-11-04

**Authors:** Eugenia Claudia Zarnescu, Narcis Octavian Zarnescu, Nicoleta Sanda, Radu Costea

**Affiliations:** 1Department of General Surgery, “Carol Davila” University of Medicine and Pharmacy, 050474 Bucharest, Romania; 2Second Department of Surgery, University Emergency Hospital Bucharest, 050098 Bucharest, Romania

**Keywords:** Clavien–Dindo classification, risk factors, right colon cancer, right colectomy, severe complications, nutrition, Charlson index, blood transfusion, surgery

## Abstract

*Background and Objectives:* This study aimed to investigate the potential risk factors for severe postoperative complications after oncologic right colectomy. *Materials and Methods*: All consecutive patients with right colon cancer who underwent right colectomy in our department between 2016 and 2021 were retrospectively included in this study. The Clavien–Dindo grading system was used to evaluate postoperative complications. Univariate and multivariate logistic regression analyses were used to investigate risk factors for postoperative severe complications. *Results*: Of the 144 patients, there were 69 males and 75 females, with a median age of 69 (IQR 60–78). Postoperative morbidity and mortality rates were 41.7% (60 patients) and 11.1% (16 patients), respectively. The anastomotic leak rate was 5.3% (7 patients). Severe postoperative complications (Clavien–Dindo grades III–V) were present in 20 patients (13.9%). Univariate analysis showed the following as risk factors for postoperative severe complications: Charlson score, lack of mechanical bowel preparation, level of preoperative proteins, blood transfusions, and degree of urgency (elective/emergency right colectomy). In the logistic binary regression, the Charlson score (OR = 1.931, 95% CI = 1.077–3.463, *p* = 0.025) and preoperative protein level (OR = 0.049, 95% CI = 0.006–0.433, *p* = 0.007) were found to be independent risk factors for postoperative severe complications. *Conclusions*: Severe complications after oncologic right colectomy are associated with a low preoperative protein level and a higher Charlson comorbidity index.

## 1. Introduction

Colorectal cancer is the third-most common cancer worldwide, representing an important public health issue [1]. Several risk factors, including aging, male gender, white race, obesity, smoking, modern dietary practices (diet high in red and processed meat, diet low in fibers and vegetables, and diet low in calcium and dairy products), alcohol abuse, and family history have been linked to the development of colorectal cancer [2,3]. Development of colorectal cancer involves numerous molecular pathways regulated by multiple molecules and genes. For instance, abnormal activation of the Wnt/β-catenin signaling pathway is a crucial factor in the development and progression of colorectal cancer [4]. Loss or inactivation of adenomatous polyposis coli (APC) results in constitutive stimulation of Wnt/β-catenin signaling, which is considered a precursor to colorectal cancer [5]. Studies suggest that the molecular pathways do not act in isolation, but are interconnected, so that changes in one lead to changes in another [6]. Understanding how signaling pathways are interconnected is essential for the development of targeted cancer therapies.

Oncologic right hemicolectomy is a common surgical operation since nearly one-third of all colorectal malignancies are situated in the right colon. The most feared complication following surgery for colorectal cancer is anastomotic leakage (AL). The rate of AL after right hemicolectomy with ileo-colic anastomosis is significantly lower than the rate of AL after left colonic resection with colorectal anastomosis. Following right colon cancer resection, the incidence of AL ranged from 0.02 to 8.8%, whereas it ranged from 2 to 20% after left colon cancer resection [7,8,9]. On the other hand, it has been demonstrated that the risk factors for AL differ between right colonic resection with ileocolic anastomosis and left colonic resection with colorectal anastomosis [10,11,12,13].

Risk factors for morbidity and mortality following oncologic right colectomy have rarely been reported, and it is not possible to stratify the risk for postoperative morbidity and mortality in patients who have undergone this surgical procedure. Knowledge of the specific risk factors for postoperative morbidity and mortality would enable an intensive postoperative follow-up in high-risk patients. Prevention can be accomplished by preparing the patient preoperatively and, if possible, correcting existing risk factors such as malnutrition or anemia prior to surgery [14]. Diverse surgical strategies, such as reinforcing the anastomosis or forming a diverting stoma to mitigate the consequences of a leak, can be used to prevent AL in high-risk patients [15]. An intensive postoperative follow-up could permit the early recognition of an AL or other postoperative complication and, consequently, the prompt initiation of treatment.

The goal of this study was to find the risk factors that are linked to severe complications after an oncologic right colectomy.

## 2. Materials and Methods

### 2.1. Study Design

From 1 January 2016 to 31 December 2021, all the consecutive patients who underwent curative-intent oncologic right colectomy at the Second Department of Surgery, University Emergency Hospital Bucharest, Romania, were retrospectively included. The surgical procedure performed was a standard right colectomy with D2 lymphadenectomy. Both elective and emergency surgical procedures were included. Patients subjected to simple ileostomy, digestive bypass, or patients below the age of 18 years were excluded from the study. Patient records and an electronic database were used to select and extract data.

### 2.2. Parameters Studied

Demographic, preoperative, operative, and short-term outcome postoperative data were noted before patient discharge. The following variables were analyzed as potential risk factors: patient’s demographic data (gender, age), tobacco and alcohol use, patient’s medical comorbidities (Charlson comorbidities index), patient’s preoperative data (ASA score, serum hemoglobin, serum creatinine), patient’s preoperative nutritional status (serum total proteins, serum albumin, obesity defined as body mass index > 30), and surgical details (localization of tumor, type of resection, type of anastomosis, hand-sewn vs. stapled anastomosis, perioperative transfusion) and postoperative data (duration of hospital stay, wound infection, anastomotic leak, reintervention, admission in the ICU, Clostridioides difficile infection, noninfectious complications). Pathological results include tumor size and characteristics, resection margins, number of involved and examined lymph nodes and TNM pathological staging.

Postoperative morbidity was defined as any complication occurring during the hospital stay or within 30 days after surgery. Wound infections were diagnosed given the presence of clear signs of inflammation on the wound margin or purulent drainage from the wound. Follow-up for infectious and noninfectious complications was carried out before hospital discharge of patients. Complications were recorded for all patients in accordance with the Clavien–Dindo classification [16]. Postoperative complications were defined as major when classified as grade III (requiring surgical, endoscopic, or radiological intervention), grade IV (life-threatening complication requiring IC/ICU management), or grade V (death of the patient), according to the Clavien–Dindo classification. The presence of AL was specifically investigated during the postoperative period. Anastomotic leakage was defined as proposed by the International Study Group of Rectal Cancer and diagnosed (1) radiologically, by computerized tomography, with the presence of intra-abdominal collection adjacent to the anastomosis; (2) clinically, with evidence of extravasation of bowel content or gas through a wound or drain; or (3) intraoperatively [17].

The study protocol was approved by the local Institutional Review Board under the number 20701/11 January 2022).

### 2.3. Statistical Analysis

All values were presented as a median and interquartile range (IQR) or proportions. Normality was assessed with the Shapiro–Wilk test. Bivariate comparisons were assessed by using the Student *t*-test (for parametric distribution) or Mann–Whitney U test (for non-parametric distribution) for continuous variables and either the Chi-square or Fisher-exact test for categorical variables. Logistic binary regression analysis was used to assess factors associated with severe complications according to the Clavien–Dindo classification. Potential variables for regression models were selected based on bivariate associations (*p* < 0.10). A two-sided *p*-value of <0.05 was considered significant. Statistical analyses were performed by using the SPSS software package, version 20 (SPSS Inc., Chicago, IL, USA).

## 3. Results

### 3.1. Patient Characteristics

Among 144 patients who underwent an oncologic right colectomy, there were 69 men (47.9%) and 75 women (52.1%), with a median age of 69 (IQR 60–78). The most frequent comorbidities were hypertension in 72 patients (50%) and diabetes mellitus in 22 patients (15.3%). Obesity (body mass index (BMI) > 30) was present in 19 patients (13.2%). The median Charlson comorbidity score was 5 (IQR 4–7). The majority of the patients had an ASA 3 score (72.7%). Preoperative lab values were hemoglobin 10.4 (IQR 9.2–11.6) g/dL, blood urea nitrogen (BUN) 35 (IQR 27–43) mg/dL, creatinine 0.8 (0.7–1) mg/dL, total proteins 6.6 (IQR 6.2–7.3) g/dL, carcinoembryonic antigen (CEA) 5.10 (IQR 2.66–14.86), and CA 19–9 13.37 (IQR 4.21–72.62).

### 3.2. Tumor

The sites of right colon cancer were cecum in 70 patients (48.6%), ascending colon in 35 patients (24.3%), and the hepatic angle and transverse colon in 39 patients (27.1%). Peritonitis was present in 14 cases (9.7%). Ten patients (6.9%) in the group undergoing emergency surgery had a bowel obstruction. Most of the patients had T3 stage (69.2%). The median lymph node retrieval was 16 (IQR 11–26). According to the TNM classification, there were 15 patients (10.4%) with stage I, 54 patients (37.5%) with stage II, 61 patients (42.4%) with stage III, and 14 patients (9.7%) with stage IV.

### 3.3. Surgical Procedures

Mechanical bowel preparation was used in 42 patients (36.5%) of those operated electively and had anastomosis. All patients received preoperative oral antibiotics. The right colectomy was performed electively in 120 patients (83.3%) and as an emergency operation in 24 patients (16.7%). Extended resection to the adjacent structures was necessary in 11.8% of patients during resection of the primary tumor. A right colectomy with ileocolonic anastomosis was performed in 133 patients (92.4%), and 11 patients (7.6%) resulted in ileostomy. Stoma was used more often (*p* = 0.001) during the emergency surgeries (6 patients, 25%) compared to the elective cases (5 patients, 4.2%). All anastomoses were performed handsewn with 138 cases side-to-side, 5 cases end-to-side, and one case end-to-end.

### 3.4. Postoperative Complications and Outcomes

Most of the patients had no complications (84 patients, 58.3%), while postoperative complications were recorded in 60 patients (41.7%). A detailed list of postoperative complications is described in Table 1. Surgical complications appeared in 27 patients (18.8%) and medical complications in 40 patients (27.8%). Seven patients had both surgical and medical complications. Anastomotic leak was diagnosed in 7 patients (5.3%), four of whom required surgical reintervention (grade C fistula, 3.7%). There is no difference (*p* = 0.952) of the fistula rate between emergency operated cases (1 patient, 5.6%) and elective cases (6 patients, 5.2%). Eight patients (5.6%) needed surgical reintervention. Out of 19 patients with diarrhea, 10 patients were diagnosed with Clostridioides difficile infection. Three patients had postoperative gastric bleeding, one requiring surgical reintervention. Seventy-nine patients (54.9%) received blood transfusions. The postoperative mortality rate was 11.1% (16 patients). Four patients died of surgical complications, all of them with surgical reintervention (anastomotic leakage grade C, intra-abdominal bleeding and 2 cases with small bowel infarction), and 12 patients died of medical complications. The mortality rate was 20.8% (5 patients) in the emergency group, compared with 9% (11 patients) in the elective group. The median length of postoperative hospital stay was 11 (IQR 10–15) days.

Complication distribution according to the Clavien–Dindo classification was grade I (12 patients, 8.3%), grade II (28 patients, 19.4%), grade III (3 patients, 2.1%), grade IV (1 patient, 0.7%), and grade V (16 patients, 11.1%).

### 3.5. Uni-Multivariable Analysis

There is a trend (*p* = 0.086) for more severe postoperative complications for patients operated on as an emergency (30%) compared to elective operations (14.5%). The impact of mechanical bowel preparation was assessed on selected 82 patients (elective cases and with ileo-colonic anastomosis) and was significantly associated with the presence of Clavien–Dindo severe complications (*p* = 0.003). The effect of blood transfusion on the use was as follows: preoperative (*p* = 0.065), intraoperative (*p* = 0.062), and postoperative (*p* = 0.005). The postoperative blood transfusion was excluded from the multivariate analysis model, this being rather the effect and not the cause of severe complications.

The factors (Table 2) that were associated with severe postoperative complications in univariate analysis (*p*-value < 0.1) were: Charlson comorbidity score, mechanical bowel preparation, level of preoperative proteins, blood transfusion (preoperative and intraoperative) and urgency (elective/emergency right colectomy). A lack of correlation (*p* > 0.05) was identified among the variables to exclude the potential issue of multicollinearity.

In the logistic binary regression, the Charlson score (OR = 1.931, 95% CI = 1.077–3.463, *p* = 0.025) and preoperative protein level (OR = 0.049, 95% CI = 0.006–0.433, *p* = 0.007) were found to be independent risk factors for postoperative severe complications.

## 4. Discussion

The present study found that nutrition status and medical comorbidities are independent risk factors for developing severe complications after oncologic right colectomy, which makes the preoperative prehabilitation the crucial step towards an uneventful period. In our series, the mortality and morbidity rates after surgery for right colon cancer were in accordance with the ranges previously reported in the literature, corresponding to 9 to 14.5% for mortality and 32 to 54.3% for morbidity [18,19,20,21].

While the most common postoperative complications were surgical site infections, the most significant surgical complication is represented by anastomotic fistula. The present study shows a rate of anastomotic leakage (5.3%) similar with that previously reported in literature, which ranged from 1.2 to 6.4% [12,13,22,23]. In a study on 1940 consecutive patients who had elective colonic resection with intraperitoneal anastomosis without a diverting stoma for colorectal adenocarcinoma, the authors concluded that anastomotic leakage occurs more frequently after colo-colic and ileo-colic anastomosis than after intraperitoneal colorectal anastomosis [24]. The right colectomy appears to be at higher risk of AL, with a greater risk of surgical intervention than after an elective left colectomy. The variation in the profile of complications between a right or left colectomy could come from the differences between anastomosis (ileo-colic vs. colo-colic) and microbiota [25].

In particular, an emergency colectomy for an obstructive tumor is associated with an increased risk of postoperative morbidity and mortality [26,27,28]. In our study, we did not find any statistical difference in terms of severe postoperative complications when comparing emergency and elective approaches, but we observed a trend for a higher rate of Clavien–Dindo III–V complications occurring in emergency surgery patients compared with the elective group (25% vs. 11.7%; *p* = 0.085). For stable patients with right-sided colon cancer obstruction, a right colectomy with primary anastomosis is the preferred option, while for unstable patients, a right colectomy with terminal ileostomy should be considered the procedure of choice [29]. In our study, stomas were used more frequently, but not exclusively, during emergency operations (25% vs. 4.2%, *p* = 0.001). Ileostomy after a right colectomy is recommended to be used in cases of suboptimal conditions. We found that anastomosis performed in emergency, with proper case selection, has a similar risk of AL compared to an elective right colectomy (5.6% in the group with emergency surgery vs. 5.2% in the group with elective surgery, *p* = 0.952). Bayar et al. evaluated the results of emergency versus elective surgery in colorectal cancer patients and found that postoperative complications were more common in the group with emergent surgery [30]. When additional diseases of the patients were evaluated, it was determined that comorbid diseases were significantly higher in the emergent group. One of the reasons for this finding may be the impossibility of controlling the state of the comorbidities in patients undergoing emergency surgery. Length of hospital stay, advanced stage on admission, postoperative complications such as surgical site infection, evisceration, and anastomosis leakage rates were higher in patients in the emergency surgery group.

Technical details of the ileo-colic anastomosis were actively investigated to understand their role in the risk of fistula development. Our preferred technique is handsewn, side-to-side anastomosis. Two meta-analyses that compared stapled versus hand-sewn anastomosis in colorectal surgery did not find any differences in terms of anastomotic fistula [23,31]. However, there is evidence that the stapled technique is a risk factor for clinically relevant fistula [9]. This finding was further supported by a recent cohort study that showed a 5.4% versus 2.4% rate of AL in the stapled and handsewn groups (*p* = 0.004). This difference was validated by multivariable analysis, which revealed a twofold increase in AL for the stapled technique [32].

Age is an important risk factor for colorectal cancer. In large case series, it was found that it peaked in the seventh decade [33,34]. In our study, the median age of patients with emergency and elective surgery was 69 years. Patients who developed severe postoperative complications were older (74 vs. 69) compared to patients with mild complications or no complications. Furthermore, in our series, patients developing severe postoperative complications had a higher Charlson score. This finding suggests that advanced age and the presence of comorbidities lead to an increase in the severity of postoperative complications. Our findings corroborate the findings of a review of 1983 patients published by the French National Surgical Association, which demonstrated that thirty-day postoperative mortality after emergency surgery for obstructing colon cancer is correlated with age, ASA score, pulmonary comorbidity, right-sided colon cancer, and hemodynamic failure [35]. A recent study of 593 cases of right-sided colon cancer resections showed that a higher Charlson comorbidity index was identified as an independent predictor of postoperative anastomotic leakage (HR 4.91, 95% CI 2.23–10.85, *p* < 0.001) [36]. Bakker et al. found that older age, high ASA grade, high Charlson score, and emergency surgery are independent risk factors for death after anastomotic leakage [13]. In this study, the risk of death after AL was twice as high following a right colectomy compared with a left colectomy. Several other studies have found higher rates of morbidity and mortality in emergent colorectal surgery compared with elective surgery [37,38]. This disparity was attributed to patients’ comorbidities as well as situations that raise surgical risks, such as hydroelectrolytic imbalances and operating on an obstructed and unprepared colon, all of which contributed to an increase in postoperative complication rates. A poor preoperative nutritional status of the patient seems to be the main risk factor for anastomotic leakage after an elective right colectomy with ileo-colic anastomosis for cancer. Previous research has shown that AL after a colorectal resection is linked to low levels of serum albumin [39] or total proteins [9].

An efficient mechanical bowel preparation (MBP) was considered to be an important factor in preventing infectious complications and anastomotic dehiscence after colorectal surgery [40]. In retrospective series, intestinal preparation with mechanical and oral antibiotics has been reported to reduce surgical site infections after a colectomy, compared to no intestinal preparation [41]. Several meta-analyses have established a beneficial effect of combined mechanical bowel preparation and preoperative oral antibiotics to reduce the incidence of surgical-site infection and potentially that of anastomotic leak, with some support from observational studies for the use of oral antibiotics alone [42]. Our study showed, in univariate analysis, fewer severe complications in patients treated with intestinal preparation. Newer reports consider that the administration of oral antibiotics as prophylaxis the day before colon surgery significantly reduces the incidence of surgical-site infections without mechanical bowel preparation and should be routinely adopted before elective colon surgery [43]. Mechanical bowel preparation with oral antibiotics is the preferred preoperative preparation strategy in elective colectomy due to the low incidence of surgical site infection and anastomotic leakage [44]. The optimal strategy for bowel preparation to minimize the risk of anastomotic leakage remains to be established [45].

Several studies determined that patients with right colon cancer had a significantly higher incidence of anemia at admission than those with left colon cancer, leading to a higher need for transfusions of blood products in these cases [46,47]. In our series, more than half of the patients required a blood transfusion, and there was a trend for an association between blood transfusion and severe postoperative complications. Some authors suggested that preoperative anemia, regardless of its severity, was independently associated with a worse prognosis after surgery in patients with colonic cancer, possibly due to an association between preoperative anemia and highly advanced tumors [46,48]. At the same time, anemia has been proven to be associated with a higher risk of anastomotic fistula, most probably through impaired anastomotic healing induced by local ischemia [49]. On the other hand, blood transfusion has also been shown to have a deleterious effect on both short-term [50] and long-term results after colorectal surgery [51]. The immunological suppression induced by blood transfusion increases the risk of infections around anastomoses. Kwon et al. state that the use of perioperative blood transfusion, without mild or severe preoperative anemia, was independently associated with worse overall and recurrence-free survival in nonmetastatic colorectal cancer. For better oncological outcomes, their findings indicate the need to reduce the use of blood transfusion in the perioperative period [51]. To avoid the liberal policies for blood transfusion, the Patient Blood Management (PBM) has been introduced, which is an evidence-based multimodal approach for blood transfusion optimization [50,52]. Figure 1 summarizes previously listed factors.

As a limitation of the present study, it must be considered the retrospective character of the study. The main issue of missing data was mitigated by rigorous data collection form medical charts and operatory note reports. Another limitation is represented by the number of patients included in this study. Despite a relatively small number of patients, we were able to demonstrate that a higher Charlson index and lower protein levels are independent risk factors for postoperative severe complications. In univariate analysis, preoperative blood transfusion and lack of mechanical bowel preparation were statistically significant predictors of severe postoperative complications. However, a larger patient sample is required to confirm these findings. 

## 5. Conclusions

Right colectomy with ileo-colic anastomosis for colon cancer is still associated with a significant rate of morbidity and mortality. Our study suggests that a higher Charlson score and poor nutritional status are independent risk factors for postoperative severe complications after oncologic right colectomy. For high-risk patients, staged surgical management including primary defunctioning stoma or resection without anastomosis might represent a safer option. Cancer prehabilitation and case selection for anastomosis/stoma are essential for an uneventful postoperative period after oncologic right colectomy.

## Figures and Tables

**Figure 1 medicina-58-01598-f001:**
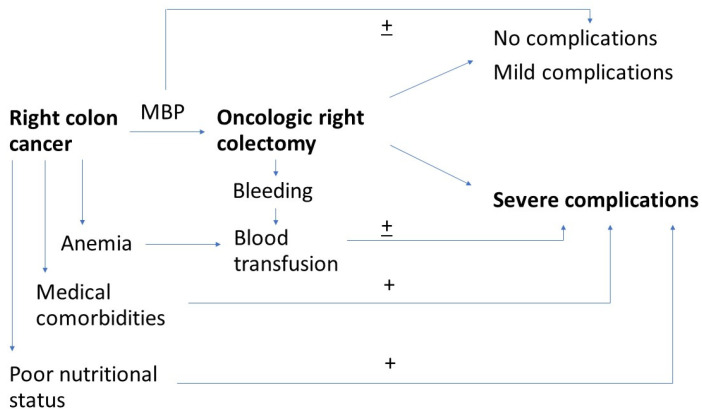
Factors associated with severe postoperative complications after oncologic right colectomy.

**Table 1 medicina-58-01598-t001:** Postoperative complications according to Clavien–Dindo grading.

Postoperative Complications	Mild Complications nr. of Complications (%) *n* = 40	Severe Complications nr. of Complications (%) *n* = 20
Surgical complications (*n* = 27)		
Wound complications	14 (35%)	1 (5%)
Anastomotic fistula	3 (7.5%)	4 (20%)
Intra-abdominal bleeding	0	1 (5%)
Gastric bleeding	0	1 (5%)
Partial bowel necrosis	0	1 (5%)
Small bowel obstruction	0	1 (5%)
Small bowel perforation	0	1 (5%)
Medical complications (*n* = 40)		
Neurologic complications	2 (5%)	0
Pulmonary complications	3 (7.5%)	2 (10%)
Cardiac complications	1 (2.5%)	8 (40%)
Renal complications	3 (7.5%)	0
Digestive complications	18 (45%)	3 (15%)

**Table 2 medicina-58-01598-t002:** Univariate analysis of prognostic factors for postoperative severe complications.

Variables	No Complications and Mild Complications *n* = 124	Severe Complications *n* = 20	*p*-Value
Gender Male Female	61 (49.2%) 63 (50.8%)	8 (40%) 12 (60%)	0.445
Age	69 (59–77)	74 (66–79)	0.061
Obesity	17 (11.8%)	2 (1.4%)	0.695
Hypertension	62 (50.4%)	10 (50%)	0.973
Diabetes mellitus	21 (16.9%)	1 (5%)	0.169
Charlson score	5 (4–7)	6 (5–9)	0.050
Mechanical bowel preparation Yes No	41 (40.27%) 61 (59.8%)	1 (7.7%) 12 (92.3%)	0.022
Hemoglobin (g/dL)	10.4 (9.2–11.6)	10.3 (9.5–11.3)	0.852
BUN (mg/dL)	35 (28–43)	28 (20–44)	0.137
Creatinine (mg/dL)	0.84 (0.7–1)	0.7 (0.6–1)	0.091
Total proteins (g/dL)	6.7 (6.32–7.43)	6.1 (5.1–6.8)	0.014
Urgency Emergency operation Elective operation	18 (14.5%) 106 (85.5%)	6 (30%) 14 (70%)	0.085
Extensive resection	15 (12.2%)	2 (10%)	0.778
Ileo-colic anastomosis Yes No	117 (94.4%) 7 (5.6%)	16 (84.2%) 4 (15.8%)	0.106
Blood transfusion (units) Preoperative Intraoperative Postoperative	1 (0–2) 0 (0–0) 1 (0–2) 0 (0–0)	3 (0–6) 0 (0–3) 0 (0–1) 0 (0–3)	0.065 0.062 0.005

Data are expressed as number of patients (%) or median (25–75th percentile). blood urea nitrogen–BUN

## Data Availability

Not applicable.

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
