# Peer review of "Risk Factors for Severe Postoperative Complications after Oncologic Right Colectomy: Unicenter Analysis"

_medicina, 2022, doi:10.3390/medicina58111598_

Round 1

Reviewer 1 Report

I would recommend to the authors include a section to mention important colon cancer signaling pathways that have been implicated in this disease such as wnt, and risk factors such as diet, age, etc. Also, incorporate a diagram or model showing the risk factors that are linked to severe complications after an oncologic right colectomy. Overall, I believe that the manuscript is an interesting short article and well written.

Author Response

Dear Reviewer 1,

Thank you very much for your comments and recommendations aiming to improve the quality of our manuscript!

Please find bellow the answers to your questions, comments and recommendations. I would recommend to the authors include a section to mention important colon cancer signaling pathways that have been implicated in this disease such as wnt, and risk factors such as diet, age, etc.

We have added to the Introduction section the following paragraph:

“Several risk factors, including aging, male gender, white race, obesity, smoking, modern dietary practices (diet high in red and processed meat, diet low in fibers and vegetables, and diet low in calcium and dairy products), alcohol abuse, and family history have been linked to the development of colorectal cancer [2,3]. Development of colorectal cancer involves numerous molecular pathways regulated by multiple molecules and genes. For instance, abnormal activation of the Wnt/ β-catenin signaling pathway is a crucial factor in the development and progression of colorectal cancer [4]. Loss or inactivation of Adenomatous polyposis coli (APC) results in constitutive stimulation of Wnt/ β-catenin signaling, which is considered a precursor to colorectal cancer [5]. Studies suggest that the molecular pathways do not act in isolation, but are interconnected, so that changes in one lead to changes in another [6]. Understanding how signaling pathways are interconnected is essential for the development of targeted cancer therapies.”

Also, incorporate a diagram or model showing the risk factors that are linked to severe complications after an oncologic right colectomy.

We attached the model (Figure 1)

MBP – mechanical bowel preparation

Once again thank you very much for your recommendations!

Sincerely yours,

Narcis Octavian Zarnescu

Reviewer 2 Report

Interesting study, very good written, perfect english. But:

1)      The design of the study is not quite clear until the last paragraph with the study limitations. Since the number of your ethic approval is 20701/11.01.2022 and your data starts from 2016 I believe it is a retrospective study which I think should be clearly stated before. And since it is retrospective I can hardly understand how and when:« Informed consent was obtained from all subjects involved in the study«

2)      You are writing about oncological right hemicolectomy. You never specify what does this term means to You. Is it a simple segmental resection? Or perhaps CME on the other side? This is very important because complications would be much more often in more complicated procedures.

3)      Line 111 Peritonitis was present in 14 cases (9.7%). I dont understand this sentence between demographic data and staging. Do you mean like emergency procedures? Than You should list obstructions too.

4)      Line 113 TNM classification 15 patients stage I, 54 patients stage II, 60 patients  stage III, and 14 patients stage IV. One is missing!

5)      Line 127 while postoperative complications were recorded in 60 patients. Surgical complications appeared at 27 patients and medical complications in 40 patients. There are 67 patients and not 60.

6)      Table 1 The numbers dont match. Mild complications N=40 but if I count them there are 44. Same with severe ones.

7)      Mortality is quite high. You had 20 severe complications and 16 of patients died. Which ones? Those with surgical or medical complications? After elective or emergency operation? This is very important.

You are writing in the discussion bellow that: »In our series, the mortality and morbidity rates after surgery for right colon cancer were in accordance with the ranges previously reported in the literature, corresponding to 9 to 14.5% for mortality and 32 to 54.3% for morbidity [13-16].« All these references are for emergency surgery and those who arent have mortality of 3%.

And you didnt find any statistical diferences betwwen emerfgency and elective surgery.

8)      How did you treat 3 patients with anastomotic leak who are in Clavien Dindo 2-mild complication group? Only antibiotic? No drainage? This is strange.

9)      Table 2 The numbers dont match. In lineAge : there are only 143 patients. Why there are only 11 patients in line: Charlson score? I think all the patients were assesed with this score. Shouldnt there be 144-x patients with No or mild complications and x patients with severe complications?

10)   The table 2  is very difficult to understand. In first few rows you have percentages in brackets and bellow there are some values on the same spot. And than percentages again.

11)   Nutritional assessment is used to determine whether a person is well nourished or malnourished. It involves the interpretation of anthropometric, biochemical (laboratory), clinical and/or dietary data. It is quite bold to say that patient has a poor  preoperative nutritional status based only on the one low blood protein measurement.

Author Response

Dear Reviewer 2,

Thank you very much for your comments and recommendations aiming to improve the quality of our manuscript!

Interesting study, very good written, perfect english. But:

1)      The design of the study is not quite clear until the last paragraph with the study limitations. Since the number of your ethic approval is 20701/11.01.2022 and your data starts from 2016 I believe it is a retrospective study which I think should be clearly stated before. And since it is retrospective I can hardly understand how and when:« Informed consent was obtained from all subjects involved in the study«

Regarding study design, our study is a retrospective, observational one. We added this aspect in the abstract and study design paragraph.

All the patients admitted to our hospital sign an informed consent regarding the use of personal and medical data for clinical trials and the educational process. We retrospectively collect the patient data after obtaining ethic approval.

2)      You are writing about oncological right hemicolectomy. You never specify what does this term means to You. Is it a simple segmental resection? Or perhaps CME on the other side? This is very important because complications would be much more often in more complicated procedures.

Oncological right hemicolectomy refers to surgical indication for right colon cancer.

3)      Line 111 Peritonitis was present in 14 cases (9.7%). I dont understand this sentence between demographic data and staging. Do you mean like emergency procedures? Than You should list obstructions too.

Patients with peritonitis due to tumor perforation were included in the emergency indication group. Obstruction was present in 10 patients included in emergency group.

4)      Line 113 TNM classification 15 patients stage I, 54 patients stage II, 60 patients  stage III, and 14 patients stage IV. One is missing!

Indeed, we missed 1 patient in stage III.

''According to the TNM classification, there were 15 patients (10.4%) stage I, 54 patients (37.5%) stage II, 61 patients (42.4%) stage III, and 14 patients (9.7%) stage IV''.

5)      Line 127 while postoperative complications were recorded in 60 patients. Surgical complications appeared at 27 patients and medical complications in 40 patients. There are 67 patients and not 60.

There were 60 patients presented postoperative complications (both, surgical and medical). Considering separately, 27 patients presented surgical complications and 40 patients medical complications. These results appeared because 7 patients presented both of complications, surgical and medical (for example, one patient present diarrhea with Clostridioides difficile and anastomotic leakage).

We added this explanation in section 3.4.

6)      Table 1 The numbers dont match. Mild complications N=40 but if I count them there are 44. Same with severe ones.

We explained at previous point the results. From 40 patients in mild complications group, 4 patients presented both surgical and medical complications. Likewise, 3 patients out of 20 in the group with severe complications presented both of complications. Totally, 7 patients with both surgical and medical complications.

7)      Mortality is quite high. You had 20 severe complications and 16 of patients died. Which ones? Those with surgical or medical complications? After elective or emergency operation? This is very important.

You are writing in the discussion bellow that: »In our series, the mortality and morbidity rates after surgery for right colon cancer were in accordance with the ranges previously reported in the literature, corresponding to 9 to 14.5% for mortality and 32 to 54.3% for morbidity [13-16].« All these references are for emergency surgery and those who arent have mortality of 3%.

And you didnt find any statistical diferences betwwen emerfgency and elective surgery.

We added explanations about mortality: patients with surgical complications versus medical complications and mortality in group of emergency surgery versus elective surgery.

‘’Four patients died of surgical complications, all of them with surgical reintervention (anastomotic leakage grade C, intraabdominal bleeding and 2 cases with small bowel infarction) and 12 patients died of medical complications. Mortality rate was 20.8% (5 patients) in emergency group, compared with 9% (11 patients) in elective group’’.

We did not do a statistical analysis regarding the mortality between emergency and elective group. We analyzed differences between groups in terms of mild versus severe complications. We specified that “we observed a trend for a higher rate of Clavien-Dindo III-V complications occurring in emergency surgery patients compared with the elective group (25% versus 11.7%; p=0.085)”.

8)      How did you treat 3 patients with anastomotic leak who are in Clavien Dindo 2-mild complication group? Only antibiotic? No drainage? This is strange.

Three patients with anastomotic leakage were treated conservatively because they have drainage placed intraoperatively, flow of fistula was low (grade A) and any postoperative active intervention was not necessary.

9)      Table 2 The numbers dont match. In lineAge : there are only 143 patients. Why there are only 11 patients in line: Charlson score? I think all the patients were assesed with this score. Shouldnt there be 144-x patients with No or mild complications and x patients with severe complications?

Table 2 refers to univariate analysis. Age, Charlson score, hemoglobin, BUN, creatinine and total proteins are continuous variables and the numbers represents the median and interquartile range of these values, not numbers of patients. Charlson score was assessed in all patients. The numbers represent the median of the score and intervals used in statistical analysis, not numbers of patients.

10)   The table 2  is very difficult to understand. In first few rows you have percentages in brackets and bellow there are some values on the same spot. And than percentages again.

In table 2 are presented categorical variables represented as proportions and continuous variables represented as median and interquartile range as we explained in the section “statistical analysis”.

This type of table appears in some articles (e.g. references 2, 4).

11)   Nutritional assessment is used to determine whether a person is well nourished or malnourished. It involves the interpretation of anthropometric, biochemical (laboratory), clinical and/or dietary data. It is quite bold to say that patient has a poor  preoperative nutritional status based only on the one low blood protein measurement.

Indeed, we evaluated only the biochemical data because we have had more missing data regarding the other parameters to evaluate nutritional status. Unfortunately, retrospective character of the study did not permit to have all necessary data.

Once again thank you very much for your recommendations!

Sincerely yours,

Narcis Octavian Zarnescu

Round 2

Reviewer 2 Report

1) You are writing about oncological right hemicolectomy. You never specify what does this term
means to You. Is it a simple segmental resection? Or perhaps CME on the other side? This is very
important because complications would be much more often in more complicated procedures.
Oncological right hemicolectomy refers to surgical indication for right colon cancer.

I would still like to hear what  do You understand under the term “Oncological right hemicolectomy”. Is it a simple segmental resection? Or perhaps CME on the other side? D1, D2 or D3 resection? Or something in the middle? Explain this in few sentences.

2) Line 111 Peritonitis was present in 14 cases (9.7%). I dont understand this sentence between
demographic data and staging. Do you mean like emergency procedures? Than You should list
obstructions too.
Patients with peritonitis due to tumor perforation were included in the emergency indication group.
Obstruction was present in 10 patients included in emergency group.

Ok, I understand but why are you pointing out just peritonitis? Use both peritonitis and obstruction or don’t mention any of them.

3) The table 2 is very difficult to understand. In first few rows you have percentages in brackets and
bellow there are some values on the same spot. And than percentages again.
In table 2 are presented categorical variables represented as proportions and continuous variables
represented as median and interquartile range as we explained in the section “statistical analysis”.
This type of table appears in some articles (e.g. references 2, 4).

Yes it is true, both articles are using this type of table but there is an explanation in both as: Results are n(%) or median(25th to 75th percentiles).

Author Response

Dear Reviewer,

Thank you for these observations. All these aspects were addressed in text accordingly.
